# Laser Ablation on Isostatic Graphite—A New Way to Create Exfoliated Graphite

**DOI:** 10.3390/ma15165474

**Published:** 2022-08-09

**Authors:** Maria Isabel Sierra-Trillo, Ralf Thomann, Ingo Krossing, Ralf Hanselmann, Rolf Mülhaupt, Yi Thomann

**Affiliations:** 1Freiburger Materialforschungszentrum, Stefan-Meier-Straße 21, D-79104 Freiburg im Breisgau, Germany; 2Institut für Anorganische und Analytische Chemie, Albert Straße 21, D-79104 Freiburg im Breisgau, Germany; 3Institut für Makromolekulare Chemie, Stefan-Meier-Straße 31, D-79104 Freiburg im Breisgau, Germany

**Keywords:** isostatic graphite, toluene intercalant, graphite exfoliation, pulsed laser ablation technique

## Abstract

In search of a new way to fabricate graphene-like materials, isostatic graphite targets were ablated using high peak power with a nanosecond-pulsed infrared laser. We conducted dry ablations in an argon atmosphere and liquid-phase ablations in the presence of a liquid medium (water or toluene). After the dry ablation, the SEM images of the target showed carbon in the form of a volcano-like grain structure, which seemed to be the result of liquid carbon ejected from the ablation center. No graphite exfoliation could be achieved using dry ablation. When using liquid phase ablation with water or toluene as a liquid medium, no traces of the formation of liquid carbon were found, but cleaner and deeper craters were observed. In particular, when using toluene as a liquid medium, typical graphite exfoliation was found. We believe that due to the extremely high pressure and high temperature induced by the laser pulses, toluene was able to intercalate into the graphite layers. Between the laser pulses, the intercalated toluene was able to flash evaporate and blow-up the graphite, which resulted in exfoliated graphite. Exfoliated graphite was found on the ablated graphite surface, as well as in the toluene medium. The ablation experiments with toluene undertaken in this study demonstrated an effective method of producing micrometer-sized graphene material. When using water as a liquid medium, no massive graphite exfoliation was observed. This meant that under the used laser conditions, toluene was a better intercalant for graphite exfoliation than water.

## 1. Introduction

Since the discovery of different carbon allotropes, which involve 0D (fullerene), 1D (carbon nanotubes) and 2D (graphene) nanostructures, carbon allotropes have steadily attracted the interest of science and industry in recent years [1,2,3,4,5,6]. One of the important methods used to produce these nanostructured materials is the laser irradiation technique [7,8,9,10]. The ablation of graphite targets was reported to produce fullerenes [1] or carbon nanotubes (CNTs) [10]. Furthermore, nanoparticles, such as Fe/Ni and Co/Ni, were added as catalysts to actively produce single-wall CNTs [11,12,13]. In contrast to CNTs and fullerenes, 2D-structured graphene-based materials are usually produced in large quantities via the exfoliation of graphite materials, where an intercalant is introduced into the graphite layers and flash evaporated via thermolysis [14,15,16,17,18]. Moreover, the use of the pulsed laser ablation in the liquid phase to exfoliate highly ordered pyrolytic graphite (HOPG) and black phosphorus (BC) was reported; however, there are very few publications [19,20,21].

Liquid media are commonly used to effectively enhance the laser ablation rate of solid materials [22]. The enhancement was reported to be caused by the formation of superheated liquids under high pressure during the laser pulse and its rapid vaporization between two pulses [23,24,25]. It was reported that a laser beam on a silicon surface covered with a water film can induce a “water hammer” or liquid jet with a pressure of up to several hundred MPa [26]. In addition, the same authors reported that plasma was generated by the laser beam within the confined liquid layer, which triggered an even higher pressure. Other authors attributed the improved ablation observed under water to the transmission of a plasma plume via a dynamic fluid flow [27], which facilitates the removal of the target material. Ren et al. suggested that when a plasma is confined in a liquid, the loss of energy is reduced and the etching is improved [28]. Thus, we illustrate the phenomenon in the following simplified way (Figure 1).

Due to its unique electro-optical properties, graphite interacts with laser pulses, absorbing even strong light energies and converting them into thermal energy more efficiently than any other semiconductor or metal material. The efficient light-to-heat energy conversion on graphite was highlighted by Zazula, who reviewed the pulsed laser irradiation on graphite and the resulting temperature–pressure phase diagrams [29]. 

Although laser ablation on graphite is known to produce different kinds of carbon allotropes, the origin of the products comes from graphite melting, evaporation and subsequent redeposition or growth of the nanocarbon materials. The graphite exfoliation during liquid-phase ablation, especially when using an organic liquid medium, remains unexplored. Additionally, most laser ablation studies are performed on HOPG. In this study, we used an isostatic type of graphite, which is made of micro-sized graphite particles and has a porous microstructure. Its structure is thus fundamentally different from that of HOPG and it can adsorb photons even more efficiently than HOGP. Apart from its microparticulate character, isostatic graphite does not have an overall layered crystal orientation like HOPG. These offer a new chance to obtain laser/graphite interaction in all crystal orientations and to produce exfoliated graphite with a sheet width in the micrometer range. The purpose of this study was to find a new method based on a pulsed laser technique to produce exfoliated graphite. Two different pathways were tested. First, the isostatic graphite targets were irradiated by the laser in an inert atmosphere (dry ablation). Second, graphite covered with a thin film of liquid medium (water or toluene) was irradiated. For simplicity, all ablation experiments were executed as single spot experiments.

## 2. Materials and Methods

### 2.1. Materials

The isostatic graphite was purchased from Mersen (Suhl, Germany) (Mersen^®^ grade 2020). This type of graphite was fabricated by pressing graphite powder at high pressures (up to 400 MPa) and high temperatures (up to 2000 °C); it had micron-sized pores (9% porosity) and a 15 µm average graphite grain size (the 3D structure was characterized using µ-CT in our lab; see Appendix A Appendix A). As stated by the manufacturer, the purity of the graphite was 99.99% and the density was 1.76 g/cm^3^. The graphite surface used in this study was created by mechanically fine drilling the block into platelets.

### 2.2. Laser Technique and Parameters

Laser ablations were carried out in a sealed reactor that was connected to a Nd:YAG nanosecond-pulsed laser source. A 1064 nm wavelength, 15 Hz repetition rate and 5 ns pulse length were applied, characterized by a long repeating time (6 × 10^7^ ns) in comparison to the very short pulse length (5 ns). Table 1 shows the laser parameters used in this work. 

Short pulses with high peak powers and a low pulse repetition frequency (long period) were able to destructively ablate the surface material but heat the surrounding area only minimally.

Table 1 lists the used laser parameters. E_p_ is the pulse energy, P_p_ is the peak pulse power and E_d_ is the pulse energy density (also known as the fluence). The parameters were calculated using the following equations:(1)$$E_{p}\;[\mathrm{mJ}]=\frac{\;\mathrm{average}\;\mathrm{power}\;\left[\mathrm{mW}\right]}{\mathrm{Frequency}\;[s^{-1}]\;}$$
(2)$$P_{p}\;[W]=\frac{\mathrm{Average}\;\mathrm{power}\;\left[\mathrm{mW}\right]}{\mathrm{Frequency}\;\left[s^{-1}\right]\times \mathrm{Pulse}\;\mathrm{duration}\;\left[\mathrm{ns}\right]}\times 10^{6}$$
(3)$$E_{d}\;[J\;{\mathrm{cm}}^{-2}]=\frac{E_{p}\left[\mathrm{mJ}\right]}{\frac{\pi }{4}{\left(d_{\mathrm{laser}\;\mathrm{spot}}\;\left[\mathrm{cm}\right]\right)}^{2}}\times 10^{-3}$$

We applied the following parameters to calculate the laser parameters: The average powers of 50, 130, 240, 360 and 490 mW were set on the laser source apparatus, corresponding to pulse energies (E_p_) of 3, 9, 16, 24 and 32 mJ, as calculated using Equation (1). The on-sample spot size (diameter d_laser spot_) was estimated to be 0.04 cm (400 µm, according to SEM images). Thereafter, the laser ablation was performed by using irradiation over 1, 3, 10, 60 (defined as the standard irradiation time) and 180 s (defined as an extended irradiation time), which were equivalent to 15, 45, 150, 900 and 2700 pulses.

Figure 2 shows the experimental setup in which the graphite targets were placed at the bottom of the reactor and a liquid medium was added to form a 0.3 mm thin film on top of the graphite surface. The reactor chamber was flooded with argon gas to prevent the oxidation of the graphite.

The typical laser pulses show a Gaussian power distribution over time with a full width at half maximum of 5 ns; thus, the thermal energy pulsed into the graphite lattice involved a period that was small enough to eliminate the sublimation processes that have been observed using the traditional heating process. In addition, the pressure generated by a nanosecond-pulse laser plays an important role in the solid–liquid phase transition of the target material. Due to the extremely short time scale, it is impossible to measure the actual temperature and pressure generated by our laser system. However, according to the pulse energy density of 2.6 J/cm^2^ used in this work, the heating effect should be comparable to that used in other pulsed laser experiments [30].

### 2.3. SEM, TEM and MALDI-TOF

The different microstructures generated were characterized using SEM, TEM and MALDI-TOF. 

SEM measurements were carried out with an FEI Quanta 250 FEG SEM (Eindhoven, The Netherlands) with an acceleration voltage of 20–30 kV and a standard ETD detector in SE mode. The graphite targets were used without further treatment. Some of the SEM sample surfaces were tilted by 45° during the measurements. 

TEM measurements were carried out with a Zeiss LEO 912 OMEGA (Oberkochen, Germany) with an accelerating voltage of 120 kV. Since the newly redeposited carbon fragments (or carbon soot) can be dispersed in solvents, the dry ablated graphite targets were washed with toluene and the resulting suspensions were measured using TEM and MALDI-TOF (BRUKER Autoflex III TOF/TOF, Bremen, Germany). 

For the second pathway, i.e., irradiation in the presence of a liquid film, the liquid media in the reactor was directly collected and measured using MALDI-TOF and TEM.

## 3. Results and Discussion

In Section 3.1, we describe the results of the dry ablation (without liquid additions), first by varying the pulse energy and the irradiation time within a standard range of values and then by using higher energies and extended irradiation times. Afterward, in Section 3.2, the results of the liquid phase ablation is discussed, including the formation of exfoliated graphene. 

### 3.1. Dry Ablation under an Argon (Ar) Atmosphere

Figure 3 shows low-magnification SEM images of the craters formed using dry laser ablation with increasing pulse energies (3 mJ, 9 mJ, 16 mJ and 24 mJ) and, for comparability, with a constant irradiation time (10 s/150 pulses). The ablated crater became deeper when the pulse energy increased, which meant that more material was removed from the reaction center. In the following, we consider three processes that are responsible for the material removal: (1) carbon melting; (2) ejection of liquid carbon from the reaction center; and (3) evaporation of liquid carbon, which occurred in different zones around the ablation center. 

It was reported that a pulse laser with a power density of 10^8^ W/cm^2^ (energy density of about 3 J/cm^2^ using a 248 nm and 30 ns pulsed laser) can create local specimen temperatures of more than 5000 K, which exceeds the melting point of graphite [30]. At ambient pressure, carbon would sublimate before it reached its melting point. Yet, pulsed laser processing creates an ultra-high local pressure of up to 100 GP by using a nanosecond pulsed laser [31]. However, the reported pulse energy used was 65 J, which is much higher than the mJ-range energies used in this study. Nonetheless, we can estimate the pressure created by a pulsed laser used in this study to reach several hundred MPa. According to the report, liquid carbon could be created by an incident laser pulse with an energy density (fluency) greater than 2 J/cm^2^ [30], which is about the energy range used in this study. The created liquid carbon would exist for more than 150 ns. A higher energy density would lead to a higher temperature and longer existence time of liquid carbon (in the order of hundreds of ns). A recent review article described the carbon phase diagram, which was studied by using laser ablation techniques [32]. In addition to the high temperatures and pressures, shock waves can also be induced by pulsed lasers, which eject the liquid carbon from the reaction center into the outer circles. Finally, the ejected carbon would partially evaporate, as the pressure is reduced outside the reaction center, and partially solidifies as the temperature of the emitted liquid carbon decreases.

According to the morphologies of the craters and their substructures created by different laser pulse energies, we can distinguish the formation of four characteristic zones on the basis of the SEM images (Figure 3).

As shown in Figure 3, zone 1 was defined as the reaction center, where the graphite was directly hit by the central part of the laser beam, and therefore, was more affected by the laser than any other zone. This zone showed a diameter of approximately 350–400 µm. Zone 2 was defined as the slopes on the rim of the craters. Zone 3 was the outer circle, which was not directly affected by the laser beam but received the ejected material from zone 1 and zone 2. Zone 4 was defined as the radiating area outside zone 3, which spread out farther away from the reaction center. These zones are discussed in the following on the basis of higher-magnification SEM images.


Zone 1


As mentioned above, zone 1 was in direct contact with the beam center. Because the highest temperatures and pressures were achieved in this zone, the laser impact led to the formation of liquid carbon and, simultaneously, the dynamic mechanical shock waves pushed the liquid carbon away from the spot to the outer zones. In this case, we suggest that the amount of molten material was affected by the laser energy/impact, the efficiency of the conversion of the laser energy to thermal energy and the irradiation time. Thereafter, with increasing pulse energy, the amount of molten carbon generated by the laser pulse became greater and, as more molten material was pushed away to the outer zones, the resulting craters became deeper. Interestingly, the diameter of zone 1 was observed to be constant over the employed pulse energy range, even when very high energies and long irradiation times were applied (see Section 3.2). The diameter was determined by the effective laser spot size, which was influenced by the laser source and its focus in our apparatus setup. 

Although most of the liquid carbon was ejected by the laser impact to the outer zones, residues of the liquid carbon in this zone could be still observed, which solidified after the laser ablation, forming a thin amorphous carbon layer.

A typical whirling structure could be observed (Appendix A in the Appendix A), indicating that liquid carbon was formed in this zone, though the majority was ejected outside this zone. In the high-magnification SEM images (Figure 4), typical rounded, dune-like structures of this material can be observed. The superstructure of the original graphite particles and the porous nature of isostatic graphite were still recognizable. For lower laser energies (3.3 mJ), even traces of the original layered graphite structure were still visible, indicating that the carbon material was not 100% molten in these areas. By using a laser energy of 24 mJ, these structures did not appear. The surface was smoother, providing evidence of a more completely molten carbon. 


Zone 2


Zone 2 corresponded to the area on the rim of the craters and was found for ablations with pulse energies of 9, 16 and 24 mJ. For the ablation with a pulse energy of 3 mJ, zone 2 was absent (see Figure 3). In zone 2, the pressures and temperatures were obviously still high. Therefore, the liquid carbon could not evaporate and was simply pushed by the shock waves and flowed toward the outer circle, forming a typical wave structure. After the laser pulse, the liquid carbon cooled quickly, but the wave structure remained (Figure 5). More SEM images of these wavy structures are shown in the Appendix A (Appendix A). Please note that these wave structures were different from the structure in Figure 4a. The original particles in isostatic graphite (Appendix A) were randomly orientated with an irregular orientation of the carbon layers. These waves had a uniform structure and orientation, indicating that they were not the remnants of the original structure of the graphite particles.

We are of the opinion that these structures resulted from the hydrodynamic movement of liquid carbon in conjunction with laser-induced shock waves. Similar wavy structures were also observed by other research groups for the ablation of HOPG [33,34]. In comparison to compact HOPG, the graphite used in this study had about 9% porosity, which may have increased the energy absorption of the dynamic shock waves. Periodic structures that result from laser ablation are mostly observed and discussed in metals and semiconductors [24,35]. Mechanisms explaining the formation of periodic surface structures were proposed, which involved the phenomenon of surface plasmon–polariton waves, capillary waves and some hydrodynamic instability in the liquid melt [35,36]. However, there is no common agreement on one theory. However, it is thought that the formation of these patterns could be affected by the laser parameters, such as wavelength, pulse length, fluency and pulse frequency; the induced plasma; the responsible shock waves and some special interactions between the incident laser materials; and the electromagnetic properties of the ablated materials,


Zone 3


Zone 3 was not directly affected by the laser beam. However, liquid carbon produced in zones 1 and 2 could be mechanically expelled to this zone by the shock waves discussed above. Due to the lower pressure in this zone, the liquid carbon partially evaporated, and at the same time, the remaining liquid carbon solidified due to the lower heat. It generated a typical volcano-like microstructure with a grain-like substructure. The porous grains varied in size and become smaller toward the outer circle (Figure 6). With extended irradiation time (see Section 3.2), we could see that the area was covered by newly solidified material. It has to be noted that the area of zone 3 became narrower as the pulse energy was increased from 9 mJ to 24 mJ (see also Appendix A), implying that more liquid carbon evaporated and less remained to solidify. 

To clarify the chemical nature of the grain structure, we used a micro Raman spectrometer. The resulting spectra showed weak graphite peaks with a big halo below the D and G bands (Appendix A), which meant that the carbon is mostly amorphous. There were no traces of a diamond structure.


Zone 4


Zone 4 covered a rather large area along the outer circle of the reaction spot (Figure 3). It is unlikely that this zone was affected by the laser beam; rather, vaporized molecular fragments of carbon were deposited here.

The very fine structures of the redeposited material could be visualized in high-magnification SEM images (Figure 7). The carbon fragments found in this zone were also collected on a carbon-coated TEM grid placed near the irradiation point. According to the TEM image (Appendix A), the size of the deposited carbon fragments was estimated to be less than 10 nm. To determine whether the carbon fragments are of fullerene character, the deposited materials were washed and gathered in toluene. Thereafter, MALDI-TOF could be applied to characterize the collected carbon soot (see the following section). In contrast to a previous fullerene study [1], no fullerene materials could be found in our study. Yet, this was in agreement with a previous study [2], where the authors stated that under an argon atmosphere, fullerenes could not be created at room temperature but could at a temperature of 1200 °C. Furthermore, a Raman spectrum was taken in this study (Appendix A). Moreover, a study with X-ray powder diffractometry showed quite weak signals of graphitic materials (Appendix A). All these analyses indicated that the carbon nanoparticles seen using TEM were mostly amorphous or disordered carbon fragments. 

### 3.2. Dry Ablation with Extended Irradiation Time and Higher Laser Energy

In order to produce more ablation material and show a greater effect of liquid carbon evaporating or being removed from a crater by the laser impact, an extremely long irradiation time of 3 min (180 s) was employed. For comparison, two irradiation energies, namely, 16 and 32 mJ, were applied in an inert atmosphere (Ar). Figure 8 shows the results. Here we found the same volcano microstructure as described in Section 3.1 that contained porous and agglomerated grains in zone 3. However, more liquid carbon that was ejected from the crater could be found, and the craters were much deeper (Figure 8a vs. Figure 3c). The grain structure in zone 3 and the deposits in zone 4 had the same origin as those observed in Section 3.1. 

Comparing Figure 8a,d, we found that the amount of ejected liquid carbon accumulated in zone 3 was different; the accumulation for the 16 mJ pulse energy was much higher than for 32 mJ. This indicates that a higher energy (32 mJ) led to more evaporation of liquid carbon and less solidification of liquid carbon, probably due to a higher temperature of the liquid carbon. Moreover, in the high energy ablation at 32 mJ, an additional radial structure could be observed (Figure 8f). Similar structures were described by Hu et al., which were formed by a self-organization process [37]. 

After ablation, the graphite targets were washed with toluene, and some carbon soot could be collected for MALDI-TOF investigation. Despite the extended ablation, the amount of carbon fragments gathered was limited (compared to those directly gathered during ablation in a liquid medium, which is discussed in the following section). Note that washing the graphite target with toluene was not able to remove the micro-sized grains in zone 3. Nonetheless, carbon fragments were found in the wash medium, and their concentration was sufficiently high to be used for MALDI-TOF analysis (Appendix A). An intense signal at 538 (*m*/*z*) was detected, but no signals from fullerenes occurred, for example, at 720 (*m*/*z*) for C60. 

### 3.3. Irradiation in a Liquid Medium and Creation of Exfoliated Graphite

Based on the dry ablation experiments, we could show the formation of liquid carbon and the evaporation and re-deposition of carbon. However, graphite exfoliation could not be achieved using dry ablation. 

In this section, we show the effects of laser ablation in the presence of a liquid film, which showed very promising results in terms of creating exfoliated graphite. The same set of laser parameters was used for the dry ablation described in Section 3.1. Briefly, graphite targets covered with a liquid film, either water or toluene, were irradiated at a pulse energy of 24 mJ for 1 min. The thickness of the liquid film was set to 0.3 mm. After laser irradiation, the liquid medium with carbon soot was removed and collected for TEM and MALDI-TOF investigations. The graphite targets were then dried and characterized using SEM. 

The craters ablated in the presence of a toluene or water film were deeper than for the dry ablation shown in Section 3.1 (see Figure 3d). As result, Figure 9a shows one crater created by using toluene as the liquid medium. Here we can see that the liquid film zone 3 was free of ejected liquid carbon and the volcano-like structures observed for the dry ablation. Instead, we found that especially by using toluene, graphite exfoliation was successfully achieved and could be observed in zone 3 (Figure 9b). Thus, based on the high temperatures and high pressures created by the laser pulses and the plasma plume (Figure 1), we believe that toluene was able to intercalate into the graphite layers and flash off when the laser was switched off between pulses and the pressure was gone, resulting in a blow-up of the graphite and the formation of exfoliated graphite. 

To compare water with toluene, Figure 10 shows the graphite ablated at a pulse energy of 24 mJ for 1 min under water (Figure 10a in zone 2; Figure 10c in zone 3) and under toluene (Figure 10b in zone 2 and Figure 10d in zone 3). The four images are representative of their respective zones and indicate that toluene was a better intercalant than water. 

As could be observed, water and toluene triggered completely different ablation results on the graphite surfaces. Apart from the exfoliated graphite, liquid carbon might have been formed during the liquid-phase ablation. However, this material seemed to be directly ejected into the hyper-heated liquid media or evaporated and re-solidified in the liquid medium. In both liquid media, we could directly collect a large amount of carbon soot so that MALDI-TOF analysis could be done on the collected carbon soot. Although the major portion of the carbon soot particles were too big for a MALDI-TOF analysis, strong signals for small carbon particles could be obtained (Appendix A). We observed signals of molecules mainly around and smaller than 720 *m*/*z* (C_60_), such as 701 *m*/*z* (possibly C_57_HO/C_54_H_5_O_3_), 684 *m*/*z* (possibly C_57_H/C_54_H_5_O_2_), 595 *m*/*z* (possibly C_48_H_4_O) and 533 *m*/*z* (possibly C_43_H_6_O/C_42_H_2_O_2)_. Just a very small peak could be detected at 720 *m*/*z* (C_60_). These signals could be assigned to thermally decomposed C_60_. However, this is just a suggestion and not certain. Nonetheless, we noticed that the peaks for carbon fragments in water indicated smaller molecules than in toluene. This meant that the carbon material created via ablation in liquid could probably interact with the liquid. 

The carbon soot collected in the liquid media was investigated using TEM (Figure 11). Exfoliated graphite or graphene-like materials with a sheet width of µm dimension were observed in toluene (Figure 11a,b). In comparison, ablation in the presence of water showed nano-sized carbon particles and only very few graphene-like sheets (Figure 11c,d). The non-uniform particle sizes shown in the TEM images could be explained by the heterogenous thermal history of the collected carbon soot.

Interestingly, carbon fragments in this dimension were also reported by another research group, where HOPG was ablated by a femtosecond pulsed laser and in the presence of a water medium. They termed the carbon fragments as “graphene quantum dots” (GQDs), which were found to have a particle size of 2–5 nm and should have the form of graphene nanosheets [34]. 

## 4. Conclusions

In this work, we showed that laser ablation on isostatic graphite covered with a thin toluene liquid layer could successfully exfoliate graphite.

Three different experimental settings were used: (a) dry laser ablation under argon, (b) laser ablation under water, and (c) laser ablation under toluene. Different energies (fluctuations) and irradiation times were used. TEM, SEM, Raman spectroscopy and MALDI-TOF were used for characterization.

The results were different for the experimental setups. For the dry ablation, the formation and movement of liquid carbon and evaporation and redeposition of carbon were observed, but with no exfoliation of the graphite. For both ablations under liquid, the effect of the laser was much stronger than for the dry ablation. Nevertheless, the results were quite different. With water as the medium, large amounts of nano-sized (2–5 nm) amorphous carbon dots formed, while in toluene, we found a ring of exfoliated graphite around the laser beam. We think that the nonpolar toluene heated to ultrahigh temperatures under the high pressure of the laser pulse was able to intercalate the graphite layers. Between laser pulses, the intercalated toluene could flash evaporate and blow-up the graphite, leading to the formation of exfoliated graphite. The use of porous isostatic graphite with its large surface area could have enhanced this effect. This new approach provides easy access to graphene-like materials and surface modifications of isostatic graphite. Fullerene, which was thought to be a possible product, was found only in such small amounts that its formation was uncertain.

## Figures and Tables

**Figure 1 materials-15-05474-f001:**
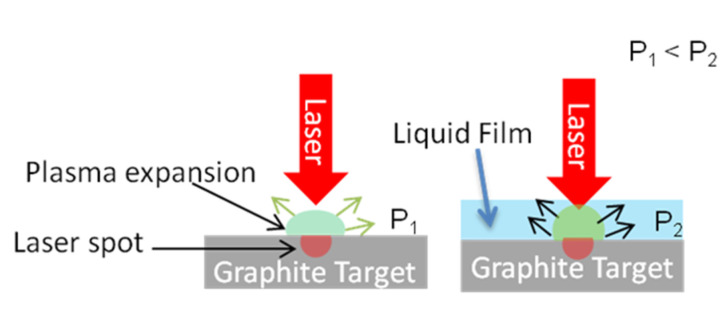
Schematics of laser ablation on dry graphite (**left**) and with a thin liquid film covering the graphite surface (**right**). Laser pulses strike the surfaces of the targets, producing plasma plumes (green), which expand into the gas (argon) atmosphere or the respective liquid, generating different local pressures (P_1_ in gas and P_2_ in liquid). In the liquid medium, the plasma is confined and causes a higher pressure than in gas: P1 < P2.

**Figure 2 materials-15-05474-f002:**
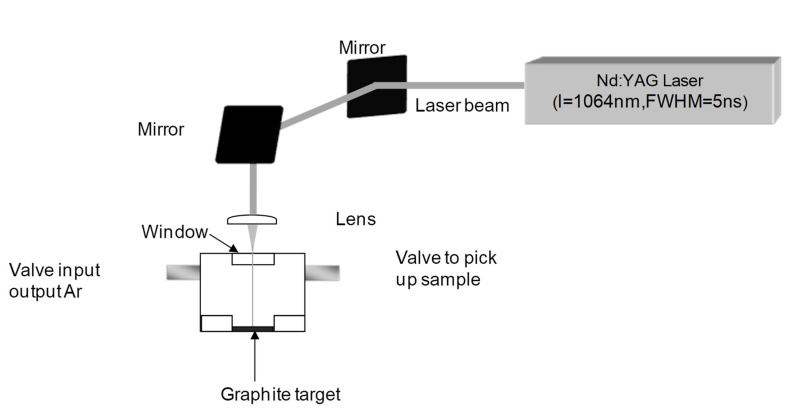
Setup of the laser apparatus and reactor. The reactor was evacuated to a pressure of 10^−2^ mbar and flooded with argon. The process was repeated several times to make the chamber as oxygen-free as possible.

**Figure 3 materials-15-05474-f003:**
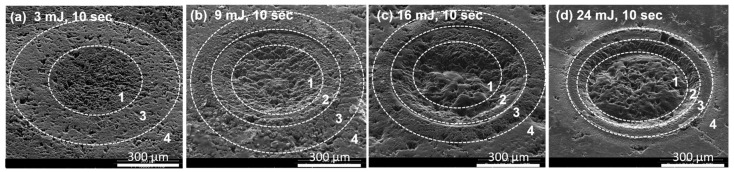
SEM images of graphite surfaces generated by different laser energies: (**a**) 3 mJ, (**b**) 9 mJ, (**c**) 16 mJ and (**d**) 24 mJ. Different zones 1–4 are indicated. The surface was tilted by 45°. Irradiation time: 10 s/150 pulses.

**Figure 4 materials-15-05474-f004:**
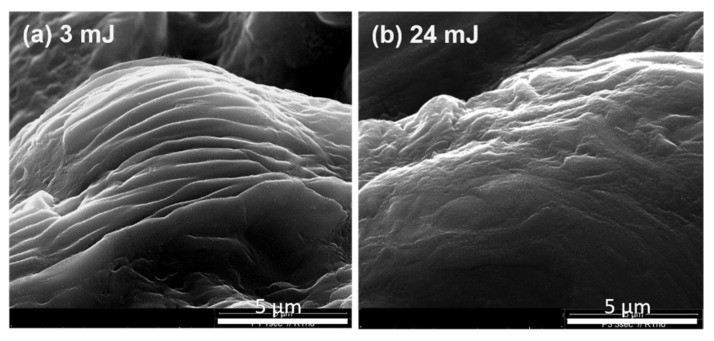
Zone 1. Structure resulting from molten graphite after ablating at two different laser energies. Irradiation time: 1 s/15 pulses.

**Figure 5 materials-15-05474-f005:**
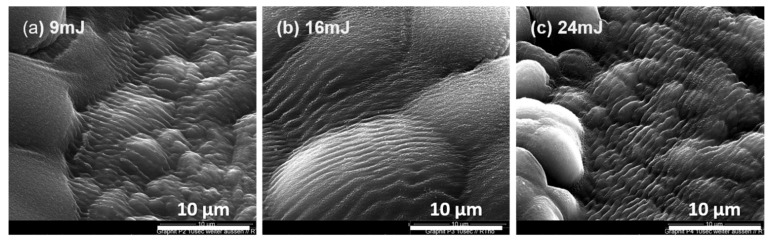
Zone 2. SEM images of graphite targets ablated with different laser energies. Irradiation time: 10 s/150 pulses.

**Figure 6 materials-15-05474-f006:**
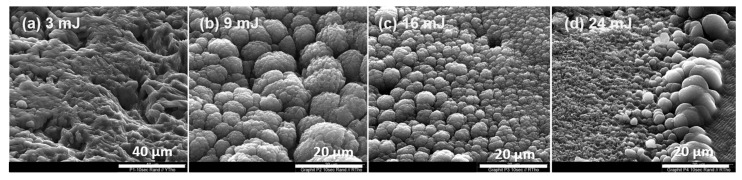
Zone 3. SEM images of graphite targets irradiated with four different pulse energies: (**a**) 3 mJ, (**b**) 9 mJ, (**c**) 16 mJ and (**d**) 24 mJ. Irradiation time: 10 s/150 pulses.

**Figure 7 materials-15-05474-f007:**
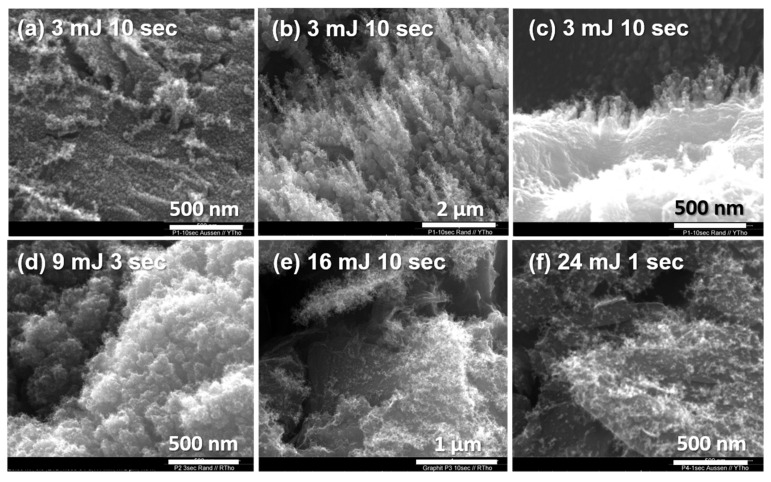
Zone 4. SEM images of deposited materials that were generated at different pulse energies and exposure times.

**Figure 8 materials-15-05474-f008:**
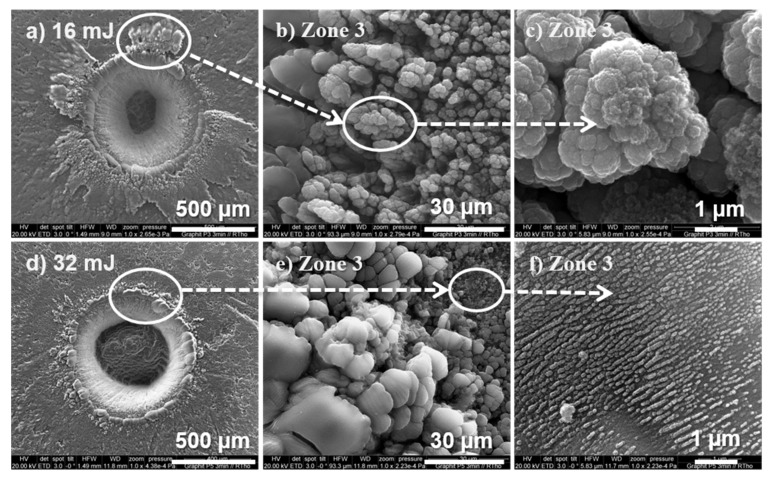
SEM images of two graphite targets irradiated over the same extended irradiation time and with different laser energies: (**a**–**c**) 16 mJ and (**d**–**f**) 32 mJ. Irradiation time: 180 s/2700 pulses.

**Figure 9 materials-15-05474-f009:**
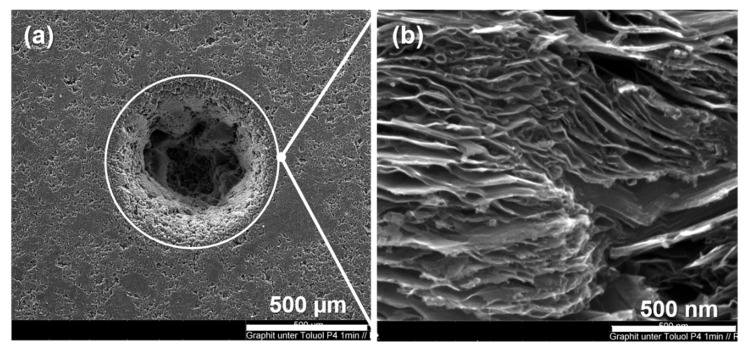
SEM images of a graphite target irradiated under a thin liquid toluene film: (**a**) the circle marks the position of zone 3 and (**b**) high-magnification SEM image within zone 3 (24 mJ, 30 s irradiation).

**Figure 10 materials-15-05474-f010:**
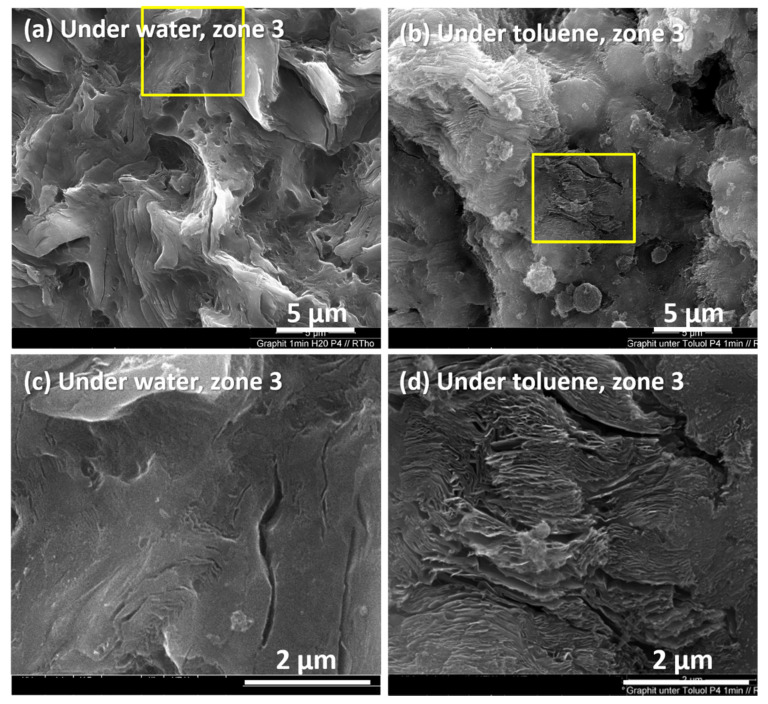
Zone 3. SEM images of graphite targets irradiated using a laser under a 0.3 mm thin liquid film: (**a**,**c**) under water and (**b**,**d**) under toluene. (**c**,**d**) are zoomed from the square frames in (**a**,**b**).

**Figure 11 materials-15-05474-f011:**
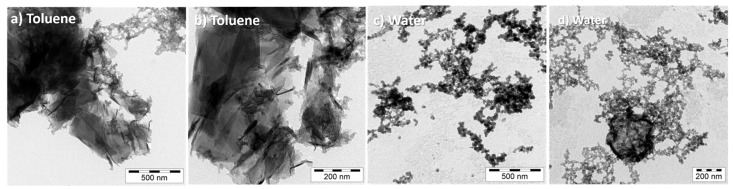
Nanostructures generated using laser ablation on a graphite target covered with a thin liquid film. Liquids: (**a**,**b**) toluene and (**c**,**d**) water.

**Table 1 materials-15-05474-t001:** Laser parameter employed in this work.

Average Power (mW)	E_p_ (mJ)	P_p_ (W)	E_d_ (J cm^−2^)
50	3.3	0.66 × 10^6^	2.6
130	8.7	1.73 × 10^6^	6.9
240	16	3.20 × 10^6^	12.7
360	24	4.80 × 10^6^	19.1
490	32	6.53 × 10^6^	26.0

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
