# Peer review of "Laser Ablation on Isostatic Graphite—A New Way to Create Exfoliated Graphite"

_materials, 2022, doi:10.3390/ma15165474_

Round 1

Reviewer 1 Report

It is a well-written article addressing an immediate industry concern. However, the article lacks depth in any of the studied parameters. The study lacks coherence and often meanders around various topics. The manuscript in the present version contains weaknesses. The interpretation of the results lacks depth, and the outcome and the conclusions are very general.

However, some aspects require clarification and modification, as detailed in the specific comments below.

1- the authors need to improve the English and should have someone review it and correct your revision before resubmitting.

2- Title

The title is probably the most crucial section of the whole paper. The title immediately clues the reader and the reviewers into what your point is and why it’s important. The title is not informative, specific, and understandable. The authors should modify the title.

3- Abstract

The abstract requires significant revision to improve the quality of the manuscript.

4- Introduction

In the introduction section, the literature review part is generally not well organized. The introduction needs to be revised.

5- The following paragraphs (sentences) are unclear:

5-1- “It was reported that the impact of a laser beam on a silicon surface covered with a water film induces the formation of a “water hammer” or liquid jet with a pressure of up to hundreds MPa [26].”

5-2- “Graphite, based on its unique electro-optical properties, interacts with laser pulses and absorbs even strong light energies, converting them into thermal energy more efficiently than any other semiconductor or metal material. This efficient conversion was highlighted by Zazula [29], who performed a study in which a graphite target was irradiated by a LHD (Large Hadron Collider) beam, with temperatures of up to 10000 K and pressures of to 100 GPa; the same author also provided a good review of pulsed laser irradiation on graphite [29].”

5-3- “The on-sample spot size (diameter dlaser spot) was estimated to be 0.04 cm (400 μm according to SEM images). Laser ablation was performed by varying the pulse energy over values of 3, 9, 16, 24 and 32 mJ (Table 1). The irradiation time was varied over values of 1, 3, 10, 60 (defined as standard irradiation time) and 180 seconds (defined as extended irradiation time), which are equivalent to 15, 45, 150, 900 and 2700 pulses.”

5-4- “As shown in Fig. 2, zone 1 is identified as the reaction center, where the graphite was directly hit by the central part of the laser beam and therefore more affected by the laser than any other zones. Zone 1 is the deepest area in the crater and shows a diameter of approximately 350 - 400 μm. Zone 2 is defined as the slopes on the rim of the craters. Zone 3 was not directly affected by the laser beam but should have received reduced heating and pressure. These zones will be discussed when considering the higher-magnification images obtained.”

5-5- “A typical whirling structure could be observed (SI-1a, b and c in Supplementary Information), indicating that liquid carbon was formed by the laser ablation process and solidified upon cooling afterward. In the high-magnification SEM images (Fig. 4), typical rounded, dune-like structures of this material can be observed.”

5-6- “Here we found the same volcano microstructure containing porous and agglomerated grains as the one with less irradiation. However, the craters are much deeper than the ones created with standard irradiation time. More liquid carbon ejected from the crater in zone 3 could be seen. In zone 3) and the deposits (in zone 4) in the outer areas of the main crater had the same origin as those observed in the zones described in section 3.1.”

6- Research significance

The purpose of the study is not mentioned in the abstract and introduction. What is the purpose of the study and the contribution of the results to the literature? Also, the significance of the study must be described in a separate section after the introduction.

7- what are the chemical constituents and physical properties of used materials?

8- The authors only hardly discuss results. Please discuss your results more deeply.

9- Conclusion

Limited new knowledge can be found in the conclusion part. The conclusion section needs to be re-written.

10- Figures

The descriptions of axis and legend in different figures are not of the same size fonts. Furthermore, authors can enhance the quality of the figures.

11- The format of some references should be checked.

Author Response

Dear reviewer,

Thank you very much for the comments and good advice. We have tried to revise the manuscript as you suggested. You can find the details below in the text marked in red.

1- the authors need to improve the English and should have someone review it and correct your revision before resubmitting.

We have corrected the English and improved it as much as we can.

2- Title

     The title is probably the most crucial section of the whole paper. The title immediately clues the reader and the reviewers into what your point is and why it’s important. The title is not informative, specific, and understandable. The authors should modify the title.

The title is now modified accordingly. “Graphite exfoliation” is included in the title, which is the main motivation of the study.

- Abstract

    The abstract requires significant revision to improve the quality of the manuscript.

Revised totally. 

4- Introduction

    In the introduction section, the literature review part is generally not well organized. The introduction needs to be revised.

We tried to highlight some references on the synthesis of carbon nanomaterials and on the laser ablation techniques. 

5- The following paragraphs (sentences) are unclear:

5-1- “It was reported that the impact of a laser beam on a silicon surface covered with a water film induces the formation of a “water hammer” or liquid jet with a pressure of up to hundreds MPa [26].”

Modified to make it clearer.

5-2- “Graphite, based on its unique electro-optical properties, interacts with laser pulses and absorbs even strong light energies, converting them into thermal energy more efficiently than any other semiconductor or metal material. This efficient conversion was highlighted by Zazula [29], who performed a study in which a graphite target was irradiated by a LHD (Large Hadron Collider) beam, with temperatures of up to 10000 K and pressures of to 100 GPa; the same author also provided a good review of pulsed laser irradiation on graphite [29].”

Text modified to avoid the confusing term “LHD”.

5-3- “The on-sample spot size (diameter dlaser spot) was estimated to be 0.04 cm (400 μm according to SEM images). Laser ablation was performed by varying the pulse energy over values of 3, 9, 16, 24 and 32 mJ (Table 1). The irradiation time was varied over values of 1, 3, 10, 60 (defined as standard irradiation time) and 180 seconds (defined as extended irradiation time), which are equivalent to 15, 45, 150, 900 and 2700 pulses.”

Modified to make the description clearer.

5-4- “As shown in Fig. 2, zone 1 is identified as the reaction center, where the graphite was directly hit by the central part of the laser beam and therefore more affected by the laser than any other zones. Zone 1 is the deepest area in the crater and shows a diameter of approximately 350 - 400 μm. Zone 2 is defined as the slopes on the rim of the craters. Zone 3 was not directly affected by the laser beam but should have received reduced heating and pressure. These zones will be discussed when considering the higher-magnification images obtained.”

Text modified.

5-5- “A typical whirling structure could be observed (SI-1a, b and c in Supplementary Information), indicating that liquid carbon was formed by the laser ablation process and solidified upon cooling afterward. In the high-magnification SEM images (Fig. 4), typical rounded, dune-like structures of this material can be observed.”

Text modified.

5-6- “Here we found the same volcano microstructure containing porous and agglomerated grains as the one with less irradiation. However, the craters are much deeper than the ones created with standard irradiation time. More liquid carbon ejected from the crater in zone 3 could be seen. In zone 3) and the deposits (in zone 4) in the outer areas of the main crater had the same origin as those observed in the zones described in section 3.1.”

Modified minorly, for better understanding.

6- Research significance

The purpose of the study is not mentioned in the abstract and introduction. What is the purpose of the study and the contribution of the results to the literature? Also, the significance of the study must be described in a separate section after the introduction.

Modified accordingly

7- what are the chemical constituents and physical properties of used materials?

We used the commercially available graphite with 99.99% carbon content.

8- The authors only hardly discuss results. Please discuss your results more deeply.

We have expanded the discussion part.

9- Conclusion

Limited new knowledge can be found in the conclusion part. The conclusion section needs to be re-written.

We have made the summary more compact and tried to highlight the crucial points.

10- Figures

The descriptions of axis and legend in different figures are not of the same size fonts. Furthermore, authors can enhance the quality of the figures.

Modified to more uniform

11- The format of some references should be checked.

Checked.

Reviewer 2 Report

Review report on the topic ‘High-peak-power nanosecond-pulse laser ablation on isostatic graphite’. Comments are listed below:

1.       Strengthen the abstract section. Abstract written very poorly. Add key findings of your work and the motive of the work in the abstract section.

2.       Discuss the motive behind the work. The clear application of the work should be discussed in the introduction section. From the introduction section application of the work is not clear.

3.       There are numerous spelling and grammatical errors. Please revise the manuscript thoroughly. Sentences are also not complete.

4.       The novelty of the work should also be discussed in a separate paragraph.

5.       Try to make a bridge between current and previously published work and specify the gap area and objective of the work. Add the specific gap observed from the literature at the end of the introduction section.

6.       Provide more detail about the experimental section.

7.       Add references for each image and table selected from other work.

8.       How was the selection of laser parameters performed?

9.       Provide the real setup used for work.

10.    Technical discussion is completely missing for SEM image.

11.    Add EDS spectra for the SE image.

12.    Surface study needs a technical discussion. In present, it looks like a technical report.

13.    Shorten the length of the conclusion section and add only key bullet points.

Author Response

Dear reviewer,

Thank you very much for the comments and good advice. We have tried to revise the manuscript as you suggested. You can find the details below in the text marked in red.

  1. Strengthen the abstract section. Abstract written very poorly. Add key findings of your work and the motive of the work in the abstract section.

Revised totally.

  1. Discuss the motive behind the work. The clear application of the work should be discussed in the introduction section. From the introduction section application of the work is not clear.

The motive of this study is to create exfoliated graphite with laser ablation techniques. We tried to make this clearee. The application of exfoliated graphite is broad and mostly well known. Originally it was intended to produce a large area field emitter for CT equipment. But this not within the scale of this publication.

  1. There are numerous spelling and grammatical errors. Please revise the manuscript thoroughly. Sentences are also not complete.

Errors corrected as well as we could.

  1. The novelty of the work should also be discussed in a separate paragraph.

We revised the conclusion totally in this line.

  1. Try to make a bridge between current and previously published work and specify the gap area and objective of the work. Add the specific gap observed from the literature at the end of the introduction section.

See in the revised manuscript lines 67 - 82.

  1. Provide more detail about the experimental section.

More detail added.

  1. Add references for each image and table selected from other work.

There are no images or tables from other work.

  1. How was the selection of laser parameters performed?

Preliminary tests have shown that there are large effects in this range.

  1. Provide the real setup used for work.

We added the type of the laser source apparatus: Continuum. More details about the real setup including the reactor can be found in the PhD thesis published in 2013.

  1. Technical discussion is completely missing for SEM image.

We enhanced the  discussion.

  1. Add EDS spectra for the SE image.

We didn’t take EDS on the graphite samples. We didn’t thought that it’s necessary for graphite samples with 99.99 % carbon content, and now the samples are unfortunately gone.

  1. Surface study needs a technical discussion. In present, it looks like a technical report.

We have discussed surface changes in the revised manuscript in the context of carbon melting, ejection, and evaporation.  

  1. Shorten the length of the conclusion section and add only key bullet points.

Modified accordingly.

Reviewer 3 Report

The manuscript describes the results of laser ablation of isostatic graphite under argon atmosphere (dry ablation), and then in the presence of water or toluene. The paper is worth considering for publication, provided the following revision points are addressed.

1)      I am confused because very often I cannot find a relationship between the figure number in the text and the images presented in the figures:

-p.7 is Fig.2 I think should be Fig.3,

-p.10 Fig.5,

-Fig. 6 is not described in the text of manuscript,

-p.12 in the text we have Fig.7a and 7d but the description of Fig 8,

-p.13 in the text we have Fig.7f but the description of Fig 8,

-after Fig.8 we have Fig. 13,

-the authors in the text refer to figure 12, which is not there.

2)      Could the authors explain the reason why toluene and water were chosen for the experiment, and why they decide to use the laser fluence 24 mJ and time 1 min?

3)      Which samples were examined by XRD?

4)      Could the authors discuss differences in results between toluene and water? Which parameters or properties could be the most significant?

5)      Could the authors comment on how the ablation parameters (fluence, time) could change the results of ablation in the presence of liquids?

Author Response

Dear reviewer,

Thank you very much for the comments and good advice. We have tried to revise the manuscript as you suggested. You can find the details below in the text marked in red.

1)      I am confused because very often I cannot find a relationship between the figure number in the text and the images presented in the figures:

-p.7 is Fig.2 I think should be Fig.3,

-p.10 Fig.5,

-Fig. 6 is not described in the text of manuscript,

-p.12 in the text we have Fig.7a and 7d but the description of Fig 8,

-p.13 in the text we have Fig.7f but the description of Fig 8,

-after Fig.8 we have Fig. 13,

-the authors in the text refer to figure 12, which is not there.

Sorry for the confusion. We renumbered the figures and checked the consistence in the text.

2)      Could the authors explain the reason why toluene and water were chosen for the experiment, and why they decide to use the laser fluence 24 mJ and time 1 min?

Water is polar and was used as standard liquid medium for liquid phase ablation.  Toluene was chosen because it is unpolar and is organic aromatic solvent and we assumed it has better compatibility with graphite than water. As for the energy: after the experiments with dry ablation, we think 24 mJ is a moderately high power and 1 min irradiation time is long enough to get an exfoliation effect. There is obviously the competing processes of carbon melting (leading to the nanoparticles) and graphite exfoliation. And there is certainly a need for further optimization. Thanks for this question.

3)      Which samples were examined by XRD?

It is a dry-ablated sample, with 24mJ and 1 min irradiation.

4)      Could the authors discuss differences in results between toluene and water? Which parameters or properties could be the most significant?

In section 3.3 we made some modification to show the difference between the water and toluene. Unfortunately, we didn’t make more experiments to compare the effects based on different energy and irradiation time.

5)      Could the authors comment on how the ablation parameters (fluence, time) could change the results of ablation in the presence of liquids?

we used only the 24 mJ energy and 60 seconds. Possibly we will get more nanoparticles with higher energy.

Round 2

Reviewer 1 Report

Authors have satisfactorily addressed my queries. 

Reviewer 2 Report

Accepted.